# Lower synaptic density is associated with depression severity and network alterations

Sophie E. Holmes [1], Dustin Scheinost[2], Sjoerd J. Finnema[1], Mika Naganawa[2], Margaret T. Davis[1], Nicole DellaGioia[1], Nabeel Nabulsi[2], David Matuskey[1,2], Gustavo A. Angarita[1], Robert H. Pietrzak[1,3], Ronald S. Duman[1], Gerard Sanacora[1], John H. Krystal[1,3], Richard E. Carson[2] & Irina Esterlis[1,3]

Synaptic loss and deficits in functional connectivity are hypothesized to contribute to symptoms associated with major depressive disorder (MDD) and post-traumatic stress disorder (PTSD). The synaptic vesicle glycoprotein 2A (SV2A) can be used to index the number of nerve terminals, an indirect estimate of synaptic density. Here, we used positron emission tomography (PET) with the SV2A radioligand [$^{11}$C]UCB-J to examine synaptic density in $n = 26$ unmedicated individuals with MDD, PTSD, or comorbid MDD/PTSD. The severity of depressive symptoms was inversely correlated with SV2A density, and individuals with high levels of depression showing lower SV2A density compared to healthy controls ($n = 21$). SV2A density was also associated with aberrant network function, as measured by magnetic resonance imaging (MRI) functional connectivity. This is the first in vivo evidence linking lower synaptic density to network alterations and symptoms of depression. Our findings provide further incentive to evaluate interventions that restore synaptic connections to treat depression.

[1] Department of Psychiatry, Yale School of Medicine, New Haven, CT 06511, USA. [2] Radiology and Biomedical Imaging, Yale School of Medicine, New Haven, CT 06511, USA. [3] U.S. Department of Veteran Affairs National Center for Posttraumatic Stress Disorder, Clinical Neurosciences Division, VA Connecticut Healthcare System, West Haven, CT 06516, USA. Correspondence and requests for materials should be addressed to I.E. (email: irina.esterlis@yale.edu)

The human brain displays remarkable neuroplasticity in response to stress. Persisting high levels of stress are thought to result in loss of synapses in circuits underlying affective and cognitive processes[1]. These reductions are presumed to contribute to the symptoms of depression associated with major depressive disorder (MDD) and post-traumatic stress disorder (PTSD)—highly comorbid disorders that may have shared neurobiological underpinnings[2,3].

Synaptic deficits are evident in MDD and might contribute to the neurobiology of PTSD. For example, post-mortem research has demonstrated lower number of synapses, and corresponding lower expression of synaptic-function-related genes in the dorsolateral prefrontal cortex (dlPFC) of individuals with MDD[4], consistent with evidence of lower levels of synaptic signaling proteins in MDD[4–6]. Preliminary evidence also points to synaptic deficits in PTSD[7,8], although post-mortem studies in PTSD are lacking. Brain imaging studies have consistently shown lower brain volume in the PFC, anterior cingulate cortex (ACC) and the hippocampus in both MDD[9,10] and PTSD[11], likely in part due to synaptic and neuronal loss[1]. Stress-related alterations in synaptic connections are also thought to underlie network dysfunction, with functional magnetic resonance imaging (fMRI) studies indicating disrupted functional connectivity within the PFC and its target limbic regions, as well as in wider brain networks in both MDD[12–14] and PTSD[15]. Stress-induced alterations in synaptic connections and resultant network dysfunction likely underlie both cognitive and affective deficits in these disorders. Preclinical studies provide further evidence of lower synaptic density in models of chronic stress and depression, specifically in PFC and hippocampal regions[16–18].

Synaptic deficits in depression and PTSD may be a target for treatment. For example, ketamine (an NMDA receptor antagonist with fast-acting antidepressant properties) rapidly increases the number and function of synaptic connections in preclinical models of chronic stress and produces rapid antidepressant-like effects[16,19]. Evidence of a functional association between synapse loss and depression is provided by studies demonstrating that inhibition of synaptic protein synthesis, resulting in a reduction in synapse number in the medial PFC, causes depressive-like behaviors in rodent models[20]. Further, depressed individuals, who exhibit deficits in cortical functional connectivity[14,21], display increased PFC connectivity during ketamine infusion[22] and 24 hours later[22,23]. Enhancing synaptic connectivity and plasticity might, therefore, restore executive control over emotions in stress disorders such as MDD and PTSD. That increasing synaptic connections reverses depressive-like behavior in animal models[19,24], and may underlie the therapeutic effects of ketamine in MDD[25,26] and PTSD[27], provides further support for the theory that synaptic loss may contribute to the neurobiology underlying depressive symptoms in these disorders, which are often resistant to conventional antidepressants.

However, synaptic density has not yet been studied in living humans with MDD and PTSD, and has, until now, been confined to small cohort post-mortem and preclinical quantification. In vivo estimates of synaptic density are now possible with the development of [11C]UCB-J—a radioligand that binds to the synaptic vesicle glycoprotein 2A (SV2A). SV2A is ubiquitously and homogenously located in synapses across the brain[28]. The number of vesicles per nerve terminal is a stable feature of neurons - the synaptic vesicle pool size is rapidly replenished during neural activity to sustain the capacity for ongoing neurotransmitter release[29]. Therefore, radioligand binding to SV2A provides an estimate of the density of nerve terminals and can serve as a proxy for the quantification of synaptic density. Our validation studies show that post-mortem quantification of SV2A is highly correlated with the commonly used synaptic density

marker synaptophysin and that in vivo binding of [11C]UCB-J PET is highly correlated with post-mortem SV2A density[30]. [11C] UCB-J also has excellent imaging properties, including high specificity, rapid kinetics, high test-retest reproducibility and the ability to produce high-quality parametric maps[30,31]. Further, PET imaging of SV2A has already been shown to be sensitive to lower synaptic density in temporal lobe epilepsy[30] and Alzheimer's disease[32]. [11C]UCB-J PET, therefore, provides a unique opportunity to investigate synaptic density across MDD and PTSD.

Given the heterogeneity of mental disorders, it is now recognized that mapping transdiagnostic symptoms to their underlying neurobiology and brain circuitry may have a greater translational impact than attempting to identify targets for categorical disorders[33]. We therefore adopted a transdiagnostic approach in an effort to move beyond categorical diagnoses and explored whether a shared neurobiological substrate, namely lower synaptic density, could underlie the symptomatic overlap between MDD and PTSD, which could help to inform a more targeted, symptom-based approach to treating these debilitating disorders. Specifically, we present results of the first known in vivo investigation of synaptic density and its association with depressive symptoms and cognitive function in a transdiagnostic sample of unmedicated individuals with MDD, PTSD and healthy comparison (HC) subjects using [11C]UCB-J and PET. Our primary hypothesis was that synaptic density, as measured by [11C]UCB-J PET, would be negatively associated with severity of depressive symptoms transdiagnostically. We examined synaptic density in the dlPFC, ACC and hippocampus based on the existing literature indicating alterations in synaptic function and decreased tissue volume in relation to chronic stress, MDD and PTSD within these regions[9–11,34–36]. Loss of synaptic connections is likely to have consequences for network organization and function. Therefore, as a secondary analysis, we examined intrinsic connectivity distribution (ICD)[37], a measure of whole-brain connectivity in regions of interest, and used follow-up seed connectivity to explore network alterations in relation to synaptic density, symptom severity, and cognitive function. We also stratified clinical subjects by depression severity (HAMD-17 score of 14; moderate severity[38]) for further group comparisons. We report the first known evidence for a relationship between synaptic density and depression in vivo. We show that lower synaptic density is associated with higher severity depressive symptoms, and that individuals with high severity symptoms exhibit lower synaptic density compared to healthy controls and individuals with low severity depression. We also demonstrate that synaptic density is associated with network alterations. Findings from this study provide further incentive to evaluate interventions that restore synaptic connections to treat depression.

## Results

**Synaptic density and depressive symptoms.** [11C]UCB-J $V_T$, indicative of SV2A (synaptic) density, was significantly negatively correlated with severity of depressive symptoms (HAMD-17) across all clinical subjects in dlPFC ($r = -0.633$, $p = 0.001$), ACC ($r = -0.634$, $p = 0.001$) and hippocampus ($r = -0.487$, $p = 0.012$; Fig. 1). All correlations survived corrections for multiple comparisons.

In an exploratory manner, we investigated associations between synaptic density and specific symptom clusters by examining correlations between [11C]UCB-J $V_T$ and four factors of the HAMD-17, including depression, anxiety, insomnia, and somatic factors[39]. The depression, somatic and insomnia factors were significantly correlated with [11C]UCB-J $V_T$ across ROIs

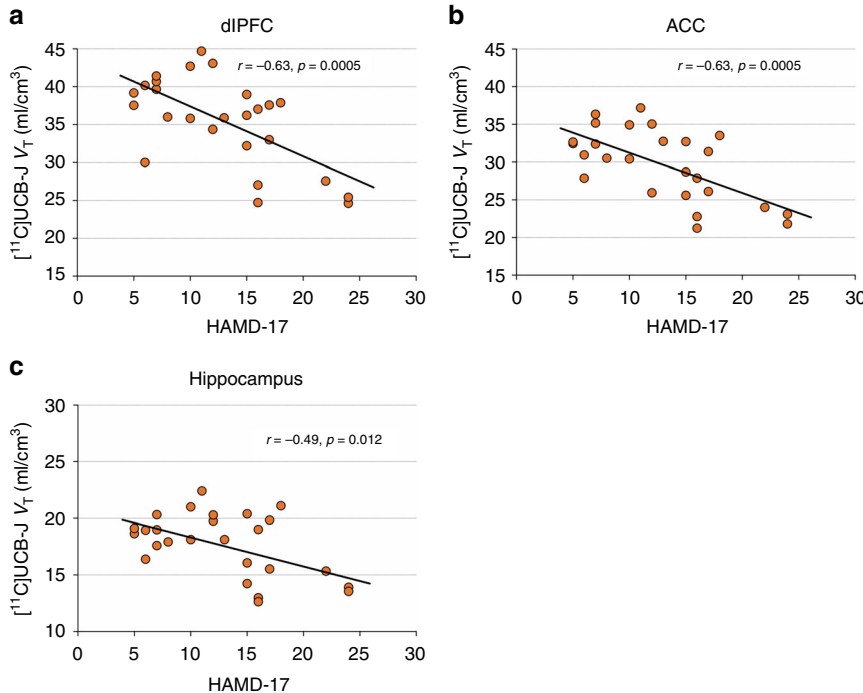

**Fig. 1** Correlations between SV2A density and severity of depressive symptoms in the full clinical sample Correlations between [$^{11}$C]UCB-J $V_T$ and HAMD-17 scores in dlPFC (**a**), ACC (**b**) and hippocampus (**c**) across all clinical subjects ($n = 26$). Correlations were computed using Pearson's $r$. dlPFC: dorsolateral prefrontal cortex, ACC: anterior cingulate cortex

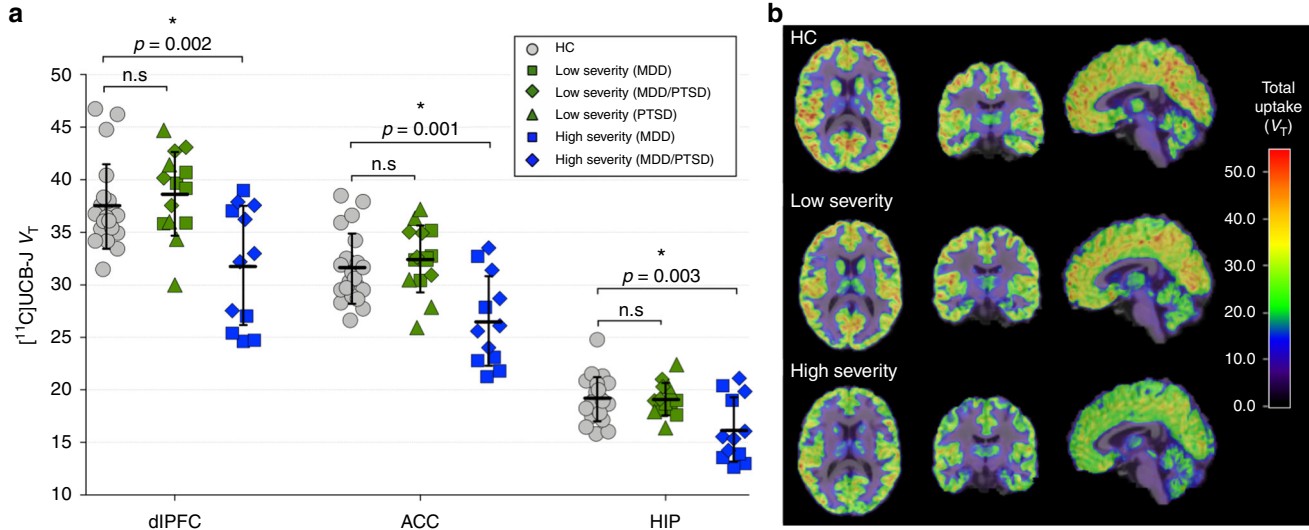

**Fig. 2** Lower SV2A density in individuals with high severity depressive symptoms vs. HC subjects. **a** [$^{11}$C]UCB-J $V_T$ in the dlPFC, ACC, and hippocampus across groups. The low severity group (in green; $n = 14$) consisted of participants with HAMD-17 scores <14 and the high severity group (in blue; $n = 12$) consisted of participants with HAMD-17 scores ≥14. Clinical subgroups are represented by different symbols; the low severity group consisted of MDD ($n = 5$), PTSD ($n = 5$) and MDD/PTSD ($n = 4$) subjects; the high severity group consisted of MDD ($n = 6$) and MDD/PTSD ($n = 6$) subjects. Group differences were assessed using MANOVA. Error bars represent standard deviation. **b** Representative axial, coronal, and sagittal parametric images of [$^{11}$C] UCB-J PET ($V_T$) scans registered to MR images in MNI space from a HC participant (top), low severity subject (middle) and high severity subject (bottom). Color bar represents $V_T$. dlPFC: dorsolateral prefrontal cortex: dlPFC, ACC:anterior cingulate cortex

(depression factor—dlPFC: $r = −0.49$, $p = 0.019$; ACC: $r = −0.43$, $p = 0.028$; hippocampus: $r = −0.40$, $p = 0.044$; somatic factor—dlPFC: $r = −0.62$, $p = 0.001$; ACC: $r = −0.60$, $p = 0.001$; hippocampus: $r = −0.45$, $p = 0.020$; insomnia factor—dlPFC: $r = −0.50$, $p = 0.009$; ACC: $r = −0.46$, $p = 0.018$; hippocampus: $r = −0.39$, $p = 0.046$). No associations were observed with the anxiety factor. There were no significant correlations between [$^{11}$C] UCB-J $V_T$ and PTSD symptoms (PCL-S; see Supplementary Notes).

**Group differences in synaptic density**. There were no significant differences in [$^{11}$C]UCB-J $V_T$ (SV2A density) between the full clinical sample ($n = 26$) and HCs ($n = 21$) across the primary ROIs (dlPFC, ACC, hippocampus) (MANOVA: $F_{3,43} = 1.88$, $p = 0.147$). In the MDD only sample ($n = 11$), a MANOVA indicated lower synaptic density across primary ROIs compared to the HC group ($F_{3,28} = 3.67$, $p = 0.024$), with the hippocampus surviving corrections for multiple comparisons (17% lower, $p = 0.002$).

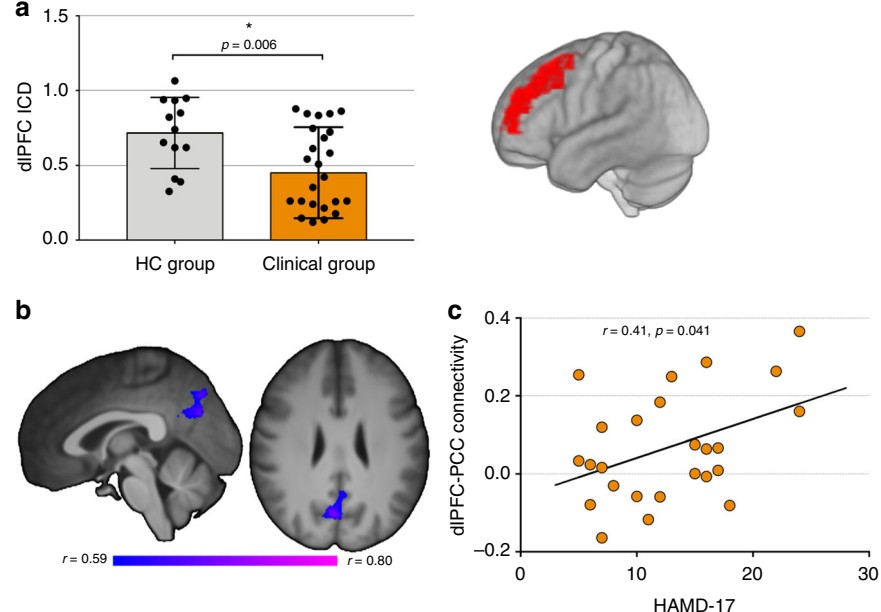

**Fig. 3** Functional connectivity and combined connectivity/PET results. **a** Significantly lower dlPFC ICD connectivity in clinical ($n = 26$) vs. HC group ($n = 13$). Error bars represent standard deviation. dlPFC region displayed on right, subsequently used as seed. **b** Negative correlation between dlPFC-PCC connectivity and dlPFC SV2A density in the clinical group. Significant voxels represent those voxels whose connectivity with the dlPFC negatively correlates with [$^{11}$C]UCB-J $V_T$ in the dlPFC. **c** Positive correlation between dlPFC-PCC connectivity and severity of depressive symptoms. Correlations were computed using Pearson's $r$. dlPFC: dorsolateral prefrontal cortex, PCC: posterior cingulate cortex, ICD: intrinsic connectivity distribution

In individuals with PTSD ($n = 15$), we observed no difference in synaptic density compared to the HC group ($F_{3,32} = 0.17$, $p = 0.917$). After stratifying clinical subjects by severity of depressive symptoms, we observed significantly lower [$^{11}$C]UCB-J $V_T$ in the high severity vs. HC group ($F_{3,28} = 5.34$, $p = 0.005$), but no difference in [$^{11}$C]UCB-J $V_T$ in the low severity vs. HC group ($F_{3,31} = 0.56$, $p = 0.647$; Fig. 2). Post-hoc tests indicated that [$^{11}$C]UCB-J $V_T$ was significantly lower in the high severity vs. HC group across ROIs. Mean [$^{11}$C]UCB-J $V_T$ was 15% lower in dlPFC ($p = 0.002$), 16% lower in ACC ($p = 0.001$), and 15% lower in hippocampus ($p = 0.003$). [$^{11}$C]UCB-J $V_T$ was also significantly different between the stratified clinical groups, with lower $V_T$ in the high severity vs. low severity group ($F_{3,22} = 5.040$, $p = 0.008$). Results of additional brain regions are shown in Supplementary Tables 1, 2 and 3, indicating that the synaptic density differences were global in nature. There were no effects of sex or smoking across groups (see Supplementary Notes).

**Functional connectivity and synaptic density**. Compared to the HC group, the clinical group displayed significantly lower ICD functional connectivity in the dlPFC ($t = 2.94$, $p < 0.05$, corrected; Fig. 3a). There were no significant differences between high and low severity groups, or correlations with symptom severity. Using the dlPFC as seed, we observed a significant negative correlation between dlPFC-posterior cingulate cortex (PCC) connectivity and SV2A density in the dlPFC in the clinical group ($r = -0.60$, $p = 0.002$; Fig. 3b; Supplementary Fig. 2). We also observed a significant positive correlation between dlPFC-PCC connectivity and severity of depressive symptoms across all clinical subjects ($r = 0.41$, $p = 0.042$; Fig. 3c). Single group seed connectivity maps are shown in Supplementary Figure 4, indicating typical patterns of connectivity from the dlPFC to other nodes of the fronto-parietal network.

**Region-to-region correlations in synaptic density**. Correlations between ACC and hippocampus [$^{11}$C]UCB-J $V_T$ were significantly stronger in the high severity clinical subjects ($r = 0.97$)

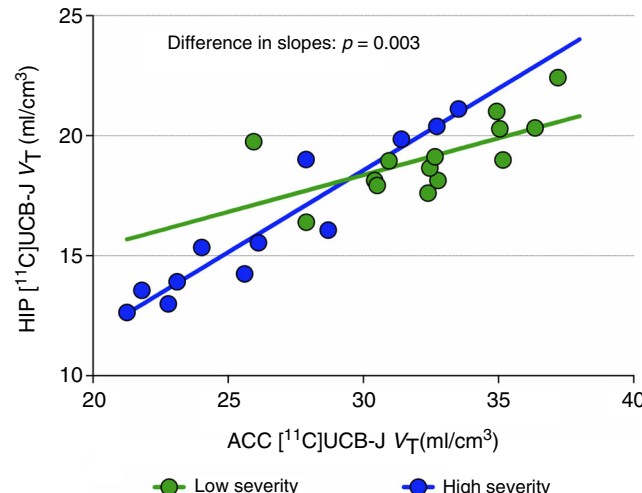

**Fig. 4** Stronger ACC-hippocampus correlation in SV2A density in high severity vs. low severity clinical groups. Differences in correlations between ROIs were assessed using Fisher's $r$ to $z$ transformation. ACC: anterior cingulate cortex, HIP: hippocampus

compared to the low severity ($r = 0.63$; $p = 0.003$; Fig. 4) and HC ($r = 0.72$; $p = 0.004$) groups. Correlations of [$^{11}$C]UCB-J $V_T$ between ACC and dlPFC, and between hippocampus and dlPFC, were not significantly different between groups.

**Cognitive function**. Clinical subjects performed significantly worse on the verbal learning (ISL) ($p = 0.041$) and verbal memory (ISL-DR) tests ($p = 0.035$) compared to HCs (Fig. 5a). High severity clinical subjects performed significantly worse on the working memory (one-back) test compared to HCs ($p = 0.044$; Fig. 5d), but low severity clinical subjects were not different from HCs. There were no between-group differences on the visual attention (IDN) test.

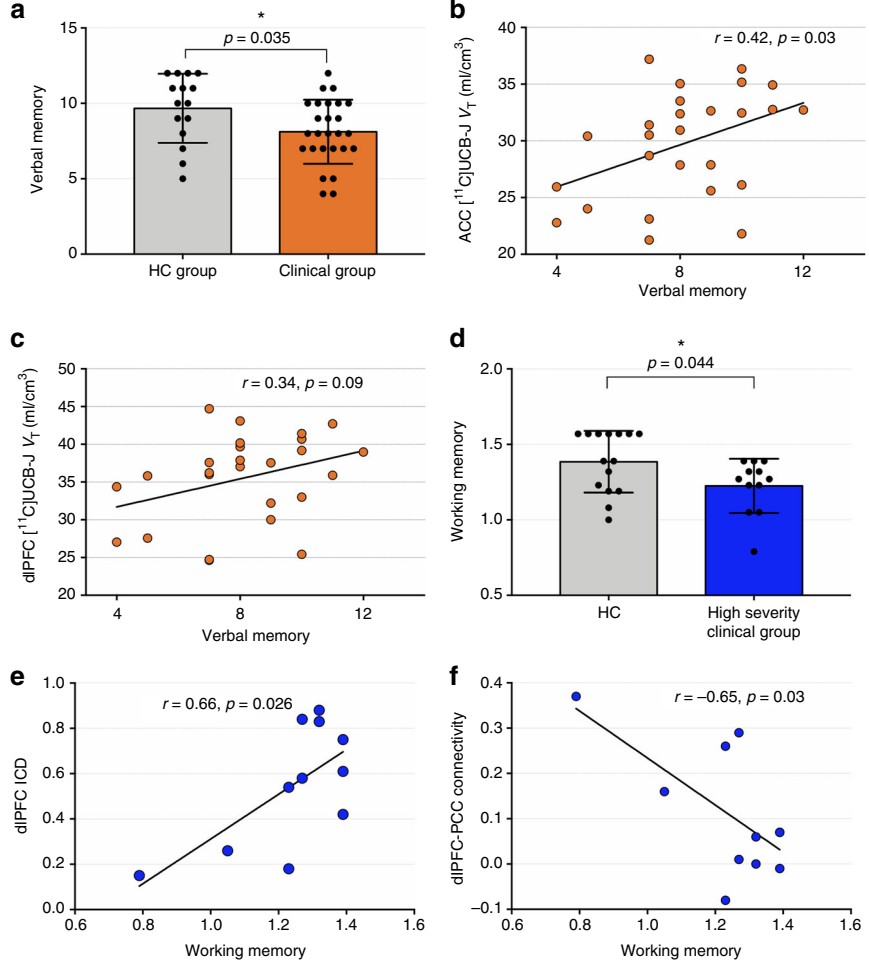

**Fig. 5** Relationship between SV2A density and functional connectivity, and cognitive function. **a** Lower performance on verbal memory task (International Shopping List-Delayed Recall) in clinical ($n = 26$) vs HC ($n = 15$) group. Error bars represent standard deviation. **b** Significant positive correlation between [$^{11}$C]UCB-J $V_T$ in verbal memory across clinical subjects. **c** Trend level correlation between dlPFC SV2A density and verbal memory. **d** Lower performance on working memory task (one-back) in high severity clinical subjects ($n = 12$) vs. HC ($n = 15$). **e** Positive correlation between dlPFC ICD connectivity and working memory in high severity clinical subjects. **f** Negative correlation between dlPFC-PCC connectivity and working memory in high severity clinical subjects, suggesting that the greater the connectivity between core nodes of two typically anticorrelated networks (default mode and central executive networks), the greater the impairment in working memory. Correlations were assessed using Pearson's $r$. dlPFC: dorsolateral prefrontal cortex, PCC: posterior cingulate cortex, ICD: intrinsic connectivity distribution

[$^{11}$C]UCB-J $V_T$ in the ACC was positively correlated with performance on the delayed verbal memory test across all clinical subjects ($r = 0.41$, $p = 0.034$; Fig. 5b), with the association between dlPFC [$^{11}$C]UCB-J $V_T$ and verbal memory scores reaching trend significance ($r = 0.34$, $p = 0.093$; Fig. 5c). There were no significant correlations with working memory or visual attention tasks, and no significant correlations in the stratified subgroups.

DlPFC-PCC connectivity was negatively associated with working memory performance across all clinical subjects ($r = -0.44$, $p = 0.030$). Examining each severity group separately, the association between dlPFC-connectivity and working memory was significant in high severity ($r = -0.65$, $p = 0.030$; Fig. 5f) but not low severity clinical subjects ($r = -0.35$, $p = 0.217$). Further, dlPFC ICD was positively correlated with working memory ($r = 0.66$, $p = 0.026$; Fig. 5e) in high but not low severity clinical groups. There were no correlations between verbal learning, memory or visual attention tasks and functional connectivity measures. These exploratory findings did not survive correction for multiple comparisons.

## Discussion

This is the first study to investigate radioligand binding to SV2A in MDD and PTSD, and the first in vivo evidence of lower synaptic density in association with depressive symptoms in these disorders. Findings suggest that lower synaptic density contributes to the severity of depression in regions associated with affective processing (dlPFC, ACC, and hippocampus), transdiagnostically. Lower synaptic density was also associated with greater severity of insomnia and somatic symptoms. Furthermore, we demonstrate lower whole-brain resting functional connectivity in the clinical as compared to HC group in the dlPFC. Using this region as seed, we found that dlPFC-PCC connectivity was negatively correlated with dlPFC synaptic density, and positively correlated with severity of depressive symptoms, suggesting that a lack of functional antagonism between hubs of two normally opposing networks may be driven by synaptic loss. We also show associations between cognitive function and alterations in synaptic density and functional connectivity measures transdiagnostically. Finally, the correlation in synaptic density between the ACC and hippocampus was greater

in the high severity compared to low severity and HC groups, suggesting that synaptic loss may occur in a coordinated fashion in those who are more severely depressed. Findings of this study therefore add to the evidence base implicating a role for synaptic density changes in symptoms associated with MDD and PTSD.

The relationship between synaptic density and depressive-like behavior has been demonstrated repeatedly in preclinical models of chronic stress[16]. In vivo MRI and post-mortem research also indicates volume[9,10,34] and synaptic[4–6] deficits in relation to depression. Research to date suggests that chronic stress causes disruption of the homeostatic mechanisms that control plasticity, resulting in destabilization and a loss of synaptic connections in mood-related circuitry[16]. A wealth of MRI studies indicate structural and functional connectivity alterations in ACC, dlPFC and hippocampus in both MDD[9,10,34,40,41] and PTSD[11,35,36,42], as well as correlations with symptom severity[41,43,44]. Further, our findings are consistent with the first report of direct quantification of synaptic density in post-mortem MDD, which demonstrated lower number of synapses in the dlPFC of individuals with MDD[4]. Preclinical models of chronic stress also confirm lower volume and synaptic density in these regions[16–18].

We hypothesized that a lower number of synapses would be associated with reduced efficacy or efficiency of functional connectivity within and between networks underlying mood and cognition, contributing to more severe symptoms. In line with this network-based hypothesis, the correlation in synaptic density between the ACC and hippocampus was greater in the high severity clinical group compared to the HC and low severity clinical groups. This suggests that synaptic deficit in each region is not independent, and that more severely depressed individuals could be 'losing' synapses in a coordinated fashion across regions. If confirmed, this would be in line with the 'network degeneration hypothesis', which is supported by evidence showing that neurodegenerative diseases are not diffuse or random, but target specific large-scale networks[45]. Indeed, while we focus on results from a priori hypothesized regions, exploratory analysis of additional brain regions points to a global effect. This is in line with the conceptualization of depression as a multi-systems disorder affecting multiple networks throughout the brain[13,46].

fMRI connectivity studies in MDD and PTSD have identified aberrant interactions or "blurring" between two typically opposing networks—the central executive network (CEN), which is anchored in the dlPFC, and the default mode network (DMN), within which the PCC is a central node[47–50]. The lack of functional antagonism between these networks is thought to reflect a bias towards internal thoughts at the expense of engagement with the external world[51]. Our results suggest that synaptic dysfunction in the dlPFC may have downstream effects on functional network organization as we observed a negative correlation between dlPFC-PCC connectivity and dlPFC synaptic density. This means that lower synaptic density was associated with a weaker anticorrelation between the CEN and DMN and suggests that the observed 'blurring' between the CEN and DMN may be driven by synaptic loss in the dlPFC. Indeed, a loss of synaptic connections in the dlPFC could reflect a reduced ability to efficiently separate the two networks, possibly through reduced top-down control from the dlPFC. Further, our results suggest that reduced dlPFC ICD connectivity and blurring of the DMN and CEN might contribute to impairments in working memory, which have consistently been linked to chronic stress[52]. Higher-order cognitive functions such as working memory are primarily subserved by PFC regions and are responsible for maintaining executive control[53]. Synaptic deficits and network dysfunction in the PFC may, therefore, contribute to the reduced top-down control over emotional processing that is thought to underlie clinical symptoms in MDD[9] and PTSD[54].

Our findings also suggest that lower synaptic density may be associated with sleep disturbance and somatic symptoms. Sleep is crucial for normal brain function, yet is frequently disturbed across psychiatric disorders, particularly in MDD[55] and PTSD[56]. The link between sleep and synaptic plasticity is well established, with convergent evidence indicating that the fundamental function of sleep is the restoration of synaptic homeostasis[57]. Disturbed sleep in MDD and PTSD could therefore be further contributing to synaptic dysfunction and resultant effects on mood and cognition, though further work is needed to unravel the causal relationship between chronic stress, sleep and synaptic function. Somatic symptoms, including fatigue, pain and appetite changes, are also prevalent in both MDD[58] and PTSD[59]. Emerging research suggests that chronic inflammation, acting in conjunction with stress, may be responsible for the somatic symptoms observed in stress-related disorders[60]. Heightened inflammation has been repeatedly demonstrated in MDD[61,62] and PTSD[63], and inflammatory processes have been shown to have profound effects on synaptic plasticity[64], raising the possibility that the observed relationship between synaptic density and somatic symptoms may be mediated, in part, by inflammation. Interestingly, while lower synaptic density corresponded with higher scores on the depression, somatic and insomnia HAM-D factors, we observed no association between synaptic density and anxiety symptoms. Further, synaptic density was not associated with severity of PTSD symptoms, suggesting that synaptic dysfunction may be specific to symptoms of depression, though this needs to be further explored in a larger sample of individuals with PTSD.

The effects of stress on the brain profoundly affect cognitive processing as well as behavior. We show that worse performance on a delayed verbal learning and memory task is associated with lower synaptic density in PFC regions, consistent with studies implicating the PFC as a crucial node supporting memory[65]. Verbal learning and memory tasks measure the capacity to simultaneously store, process and retrieve relevant verbal information - processes which are subserved by a network centered around the PFC and medial temporal lobe[66]. Specifically, the PFC is thought to self-organize its own mnemonic codes that serve as retrieval cues. Stress-induced synaptic dysfunction may, therefore, contribute to impairments in verbal learning and memory that have been repeatedly observed in MDD[67] and PTSD[65].

There are several limitations to the current study. First, our use of $V_T$ as the PET outcome measure must be taken into account. The preferred outcome measure in PET studies is often the binding potential ($BP_{ND}$), calculated using a reference region that is known to be devoid of specific binding. In previously reported validation studies, we established that the centrum semiovale, a white matter rich region, contained negligible levels of SV2A. However, given the white matter pathology observed in MDD[68] and PTSD[69], the use of a white matter reference region for the current study would be sub-optimal. Indeed, we observed significantly lower $V_T$ in the centrum semiovale of the high severity compared to HC group (figure S4). Therefore, we used $V_T$, calculated using a metabolite-corrected arterial input function as previously validated[70], as our primary outcome measure. Second, the injected radioactivity dose was lower in the high severity clinical group compared to HCs. However, this was comparable to previously published [$^{11}$C]UCB-J studies[30,32,70], and both groups were within a dose range to produce optimal count-rate and image quality. The injected mass dose was also lower in the high severity clinical group compared to HCs. However, based on our previously published study in non-human primates[31], the injected mass dose was expected to have produced <1% occupancy across groups. Any effects of lower mass in the higher severity group would have resulted in a slight underestimation in

the difference in SV2A $V_T$ compared to HCs. However, as the radioligand was injected in trace amounts in all subjects, and we observed no correlation between injected mass dose and $V_T$ (Supplementary Figure 1), any effect of mass would have been minimal. Third, it must be noted that the sample sizes of each diagnostic category alone are relatively small and no PTSD subjects without comorbid MDD were in the high severity group. MDD comorbidity in a PTSD diagnosis is a sign of more severe PTSD[71]. Thus, it is expected that the PTSD group without MDD would not be in the high severity sample. Whilst our data point to a shared neurobiology in terms of depression, there are also distinct neurobiological differences between MDD and PTSD, and larger studies investigating synaptic density in distinct categorical studies are necessary. Fourth, given that SV2A is located ubiquitously on synaptic vesicles, we cannot rule out the effects of synaptic vesicle function on our findings. However, evidence suggests that the synaptic vesicle pool is stable across neurons and that synaptic vesicles form in line with synapses[29], such that we are likely capturing the association between depression and synaptic density as opposed to number of synaptic vesicles. Further, because [11C]UCB-J PET is sensitive to the presynaptic but not postsynaptic compartment, any loss of synapses related to dendritic spine atrophy may not be detectable by this measure. Hence, we could be underestimating the degree of synaptic loss in relation to depression. Finally, we did not have resting-state data for all of the HCs, possibly limiting our power to detect between-group differences in functional connectivity between high and low severity clinical groups.

This work has important implications for the development of more effective and targeted treatments for symptoms of depression. Mounting research indicates that the mechanism underlying the rapid-acting antidepressant effects of drugs such as ketamine and scopolamine involves fast changes in synaptic function and plasticity[1]. For example, ketamine and other NMDA receptor antagonists increase mechanistic target of rapamycin complex 1 (mTORC1) signaling, leading to increases in synaptic density in the PFC, and an associated reversal of depressive-like behavior in preclinical studies[1,19]. Induction of mTORC1 signaling and synaptogenesis could, therefore, reverse the loss of PFC synaptic connections that appear to underlie depressive symptoms secondary to chronic stress. The restoration of synaptic connectivity in the PFC could, in turn, reinstate inhibitory control over networks responsible for emotion and cognition, contributing to the rapid alleviation of symptoms seen in MDD[25,26] and PTSD subjects[27] treated with ketamine.

In summary, we present findings from the first known in vivo investigation of synaptic density in psychiatric subjects.

We demonstrate a relationship between synaptic density and depression severity in a transdiagnostic sample of unmedicated MDD and PTSD individuals. The combination of [11C]UCB-J PET and functional connectivity methods provides novel insight into both the molecular and network-level underpinnings of depressive symptoms in MDD and PTSD. Indeed, we demonstrate that dysfunction of a well-known circuit in depression is associated with synaptic density changes, providing a possible molecular explanation for network dysfunction in relation to symptoms of depression. Exploratory data also suggest that synaptic density and network-level alterations may underlie cognitive impairments in MDD and PTSD. Results of this study provide further incentive to discover and evaluate new treatments that increase synaptic connections and reverse the loss of synapses caused by stress. An important line of future work is to investigate the synaptogenic effects of rapid-acting antidepressants such as ketamine in humans in vivo, as well as alternative treatments that may increase synaptic connections and have stabilizing network effects, such as exercise[72] and mindfulness[73]. In conclusion, our novel in vivo findings add to the growing evidence base implicating synaptic loss in the pathophysiology of stress-related disorders, which holds promise for a new generation of fast-acting and effective treatments targeting synaptic function.

## Methods

**Participants**. Twenty-six unmedicated clinical subjects (mean age ± SD = 39.2 ± 12.1 years; 10 females) and twenty-one age, sex, and smoking-matched HC subjects (mean age ± SD = 44.5 ± 15.4 years; 8 females) participated in the study. Clinical subjects were medication-free for at least 4 months. The full clinical group consisted of 11 individuals with MDD, 5 with PTSD and 10 with comorbid MDD and PTSD. Participants were stratified based on severity of depressive symptoms using a HAMD-17 cut-off score of 14 (indicative of 'moderate severity' depression)[38]. Individuals with HAMD-17 scores <14 were classified into a group termed 'low severity' group and individuals with HAMD-17 scores ≥14 were classified as the 'high severity' group. Twelve of the clinical subjects had moderate-to-severe symptom severity (6 MDD, 6 comorbid MDD, and PTSD). Fourteen of the clinical subjects had HAMD-17 scores <14 (5 MDD, 5 PTSD, 4 comorbid MDD, and PTSD). Demographic and clinical characteristics of the clinical and HC groups are shown in Table 1.

Diagnosis was confirmed at screening (1 week or less prior to PET) using the Structured Clinical Interview for DSM-5[74]. All participants with MDD or comorbid MDD were in a major depressive episode (MDE) at screening. Depressive symptoms were additionally assessed using the Hamilton Depression Rating Scale (HAMD-17)[38] and the Montgomery and Asberg Depression Rating Scale (MADRS)[75] at screening and scan day. Exclusion criteria were lifetime history of bipolar disorder or schizophrenia; diagnosis of substance use disorder, except for tobacco use disorder, in the past 12 months; positive urine toxicology or pregnancy tests before any scan; psychotropic medication use within the past 2 months; history of loss of consciousness for more than 5 min; significant medical condition; and contraindications to MRI or PET. Exclusion criteria were the same

**Table 1 Demographic, clinical and radioligand characteristics for each group**

|  | HC group (n = 21) | Low severity clinical group (n = 14) | p-value (Low severity vs. HC) | High severity clinical group (n = 12) | p-value (High severity vs. HC) |
|---|---|---|---|---|---|
| Age (yrs) | 44.48 (15.41) | 39.14 (11.18) | 0.27 | 38.67 (13.89) | 0.29 |
| Sex (m:f) | 13:8 | 9:5 | 0.89 | 7:5 | 0.84 |
| Smokers | 4 | 4 | 0.51 | 2 | 0.87 |
| HAMD-17 | 0.27 (0.59) | 8.50 (2.77) | <0.001 | 17.92 (3.42) | <0.001 |
| MADRS | 0.47 (1.13) | 11.6 (5.79) | <0.001 | 21.75 (5.64) | <0.001 |
| Diagnoses | - | 5 MDD, 5 PTSD, 4 MDD/PTSD | - | 6 MDD 6 PTSD/MDD | - |
| Age at onset (yrs) | - | 22.25 (10.75) | - | 21.45 (10.31) | - |
| Duration of illness (yrs) | - | 16.92 (11.70) | - | 19.61 (12.38) | - |
| Injected dose (MBq) | 585.3 (151.4) | 509.5 (196.9) | 0.34 | 450.0 (156.5) | 0.02 |
| Injected mass (ng/kg) | 25.4 (12.6) | 16.5 (12.2) | 0.04 | 12.5 (11.0) | 0.007 |
| Free fraction ($f_p$) | 0.28 (0.03) | 0.27 (0.02) | 0.44 | 0.27 (0.03) | 0.48 |

p-values were computed using independent-samples t-tests.
Values are presented as mean (SD)

for the HC group, except for the addition of no current, history of, or first-degree family history of any DSM-5 diagnosis, not including tobacco use disorder. The Yale University Human Investigation Committee and the Radioactive Drug Research Committee approved the study. All participants provided written informed consent before inclusion in the study.

Participants underwent physical and neurological examination to rule out the presence of an active medical or neurological illness. Screening involved electrocardiography, hematology, blood chemistries (electrolytes, tests of thyroid, liver, and kidney function), urinalysis and urine toxicology screening, and plasma pregnancy tests (for women). To examine the association between synaptic density and specific symptom clusters, we combined HAMD-17 items based on a meta-analysis of factor structures[39] to generate scores on 4 symptom dimensions: depression, anxiety, somatic symptoms and insomnia. Participants also completed a brief computerized cognitive testing battery (Cogstate; (https://Cogstate.com/computerized-tests), consisting of the Identification Test (IDN; measuring visual attention), the One-back Test (measuring working memory), the International Shopping List (ISL) and the International Shopping List Delayed Recall (ISL-DR; measuring verbal learning and memory).

**MRI scanning.** T1-weighted MRI scans were acquired on 3-Tesla Siemens Prisma scanner. A high resolution, three-dimensional magnetization prepared rapid acquisition gradient echo (MPRAGE) T1-weighted sequence was used to exclude structural abnormality and for co-registration with PET images (TR = 1500 ms, TE = 2.83 ms, FOV = 256 × 256 mm², matrix = 256 × 256 mm², slice thickness = 1.0 mm without gap, 160 slices, voxel size 1.0 × 1.0 × 1.0 mm³). A 2D T1-weighted image with the same slice prescription as the functional images was also collected for purposes of registration. Functional runs included 340 whole-brain volumes acquired using a multiband echo-planar imaging sequence with the following parameters: TR = 1 s, TE = 30 ms, flip angle = 62°, matrix = 84 × 84, in-plane resolution = 2.5 mm², 51 axial-oblique slices parallel to the ac–pc line, slice thickness = 2.5, multiband = 3, acceleration factor = 2. All participants had two resting-state scans. Functional MRI data was not collected for all HC subjects: data are available for 13 out of the 21 HCs (6 women; mean ± SD age 42 ± 15 yrs).

**PET scanning.** [¹¹C]UCB-J was synthesized onsite and administered intravenously as a bolus over 1 min using an automated infusion pump (Harvard PHD 22/2000, Harvard Apparatus). Details of radiotracer synthesis are outlined in Supplementary Methods. Injected radioactivity dose was lower in the high severity group (450 ± 156 MBq) vs HCs (585 ± 151 Mbq; $p = 0.02$), however, both groups were within a dose range that produced good count rates and image quality. Injected mass dose was lower in high severity subjects (13 ± 11 ng/kg) vs HCs (25 ± 13 ng/kg; $p = 0.007$). Based on the in vivo affinity of UCB-J being previously determined as 3.4 nM (20 mg/kg) in non-human primates[31], the mass dose was expected to produce <1% occupancy in both groups, such that the radioligand was injected at tracer dose levels. Further, there was no correlation between injected mass dose and $V_T$ across subjects (Supplementary Fig. 1). There were no significant between-group differences in plasma free fraction ($f_p$) (Table 1), plasma parent fraction, or metabolie-corrected input function (Supplementary Table 4). Subjects were scanned on a high-resolution human brain PET camera, the High-Resolution Research Tomograph (HRRT). All PET imaging and measurement of the metabolite-corrected arterial input function was performed according to previously described procedures[30,70], and outlined in Supplementary Methods.

**PET image analysis.** The primary outcome measure was total volume of distribution ($V_T$), computed parametrically using the 1 tissue (1T) compartment model and a metabolite-corrected arterial input function, as validated previously[70]. Distribution volume ($V_T$) is the tissue-to-plasma concentration ratio at equilibrium and reflects total uptake (specific plus nonspecific binding) of the radioligand. Radioactivity in the blood volume contributes to $V_T$, however, we previously showed that inclusion of the blood volume fraction into the 1T model fit did not affect regional K1 or $V_T$[70], likely because [¹¹C]UCB-J uptake is high in brain tissue, and the vascular activity is very low in comparison. We have previously shown that the test-retest reproducibility of [¹¹C]UCB-J $V_T$ is exceptionally good and that correcting for $f_p$ worsened absolute test retest variability (aTRV) and intraclass correlation coefficient (ICC). Therefore, we used [¹¹C]UCB-J $V_T$ as the primary outcome measure but report $V_T$ normalized by $f_p$ results in the Supplementary Table 2 for completeness. Regions of interest were derived from the Automated Anatomical Labeling (AAL) atlas and applied to the parametric images using the combined transformations from template to PET space. Partial volume correction (PVC) was used to correct for effects of tissue atrophy affecting PET outcome measures. The PET data were corrected for partial volume effects using a segmented grey matter mask and the Muller-Gartner method[76]. PET data without PVC are also presented in Supplementary Table 3.

**MRI analysis.** First, images were skull stripped using FSL (https://fsl.fmrib.ox.ac.uk/fsl/) and any remaining non-brain tissue was manually removed. All further analyses were performed using BioImage Suite[77] unless otherwise specified. Images were aligned to MNI space using a 12-parameter affine registration by maximizing the normalized mutual information between individual scans and the MNI

template brain. These aligned images were averaged together to form the initial template for non-linear registration. Images were non-linearly registered to an evolving group average template in an iterative fashion[78]. A total of 4 iterations were performed with decreasing control point spacings of 15 mm, 10 mm, 5 mm, and 2.5 mm. To help prevent local minimums during optimization, a multi-resolution approach was used and 3 resolution levels were used at each iteration.

The first ten volumes of each functional run were discarded to allow for the magnetization to reach a steady state. Motion correction was performed using SPM8 (http://www.fil.ion.ucl.ac.uk/spm/). Images were warped into common space using the non-linear transformation described above using cubic interpolation and were iteratively smoothed until the smoothness of any image had a full-width half maximum of approximately 5 mm[79] using AFNI's 3dBlurToFWHM (http://afni.nimh.nih.gov/afni/). This iterative smoothing reduces motion-related confounds. All further analyses were performed using BioImage Suite. Several covariates were regressed from the data including linear and quadratic drifts, mean cerebral-spinal-fluid (CSF) signal, mean white matter signal, and mean grey matter signal. For additional control of possible motion-related confounds, a 24-parameter motion model (including six rigid-body motion parameters, six temporal derivatives, and these terms squared) was regressed from the data. The data were temporally smoothed with a Gaussian filter (approximate cut-off frequency = 0.12 Hz). A canonical grey matter mask defined in common space was applied to the data, so only voxels in the grey matter were used in further calculations. Finally, for each participant, all preprocessed resting-state runs were variance normalized and concatenated.

After preprocessing, intrinsic functional connectivity of each voxel, as measured by the ICD[37], was calculated for each subject. ICD involves correlating the time series for any voxel with every other time series in the brain or brain hemisphere. A summary statistic is calculated by modeling the entire distribution of correlation thresholds using a Weibull distribution. Specifically, the time series for any grey matter voxel was correlated with every other voxel in the grey matter. A histogram of these correlations was constructed to estimate the distribution of connections to the current voxel. The corresponding survival function was fitted with a stretched exponential with unknown variance. As variance controls the spread of the distribution of connections, a larger variance indicates a greater number of high correlation connections. Finally, this process is repeated for all voxels in the grey matter, resulting in a whole-brain parametric image summarizing the connectivity of each tissue element as the variance of the Weibull distribution.

Follow-up seed analysis was performed to explore (post-hoc) the nodes identified by ICD analysis (dlPFC) to determine the specific connections that were most responsible for group differences in ICD. The time series of the seed region in a given participant was then computed as the average time series across all voxels in the seed region. This time series was correlated with the time series for every other voxel in the grey matter to create a map of $r$-values, reflecting seed-to-whole-brain connectivity. These r-values were transformed to z-values using Fisher's transform yielding one map for each participant representing the strength of correlation to the seed region.

As group differences in motion have been shown to confound connectivity studies, we calculated the average frame-to-frame displacement for each participant's data. In line with current reports, participants with an average frame-to-frame displacement greater than 0.20 mm for any run were removed from the analysis (1 HC). There were no significant differences for motion between the clinical and HC groups (clinical group 0.09 ± 0.02 mm, HC: 0.08 ± 0.04 mm, $p = 0.26$).

**Data analysis.** Statistical analysis was performed in SPSS v22 (IBM). Group differences in demographic, clinical and radiotracer characteristics were assessed using independent-samples $t$-tests and chi-square tests. To evaluate associations between [¹¹C]UCB-J $V_T$ and clinical and demographic variables, which were normally distributed based on Shapiro–Wilk tests, we computed Pearson's $r$. Group differences in [¹¹C]UCB-J $V_T$ were assessed in the primary ROIs (dlPFC, ACC, hippocampus) using multivariate analysis of variance (MANOVA). Homogeneity of variances was checked by Levene's test. Pairwise comparisons were subsequently carried out to determine regional $V_T$ differences, corrected for multiple comparisons. As secondary analyses, we examined the association between [¹¹C]UCB-J $V_T$ and measures of cognitive function (verbal learning and memory, attention and working memory), and between-group differences in correlations in [¹¹C]UCB-J $V_T$ between ROIs. Between-group differences in cognitive function were assessed using independent-samples $t$-tests. Between-group differences in correlations between ROIs were assessed using Fisher's $r$ to $z$ transformation. For primary PET analyses (transdiagnostic correlations with depressive symptoms and group differences), tests were 2-tailed and findings were considered significant at the $p < 0.05$ level, subsequently corrected for multiple comparisons. Secondary analyses were not corrected for multiple comparisons. The MRI results are presented as secondary due to their exploratory nature. ICD values were averaged over the three a priori regions of interest (ACC, dlPFC, and hippocampus). Independent $t$-tests were used to compare the ICD data between the study groups. Significance was assessed at the $p < 0.05$ level, corrected for multiple comparisons using Bonferroni Correction. Voxel-wise correlations were used to associate the seed connectivity data from the dlPFC to both synaptic density and depressive symptoms. Imaging results are shown at a cluster-level threshold of $p < 0.05$ with family-wise error (FWE) correction as determined by AFNI's 3dClustSim program (version 16.0.09)

using a cluster-forming threshold of $p = 0.001$, 10,000 iterations, a grey matter mask, and a smoothness estimated from the residuals using 3dFWHMx.

## Data availability

The data generated and analyzed during the current study are available from the corresponding author on reasonable request. The source data underlying Figs. 1–5, S1–3 are provided as a Source Data file.

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

## Acknowledgements

We thank the staffs at the Yale PET Center, the National Center for PTSD (West Haven Campus), and the individuals who took part in the study. We also thank UCB for providing the [11C]UCB-J radiolabeling precursor and the unlabeled reference standard. Funding support was provided by the Veterans Affairs National Center for PTSD (R.H.P., R.S.D., J.H.K., and I.E.), the Nancy Taylor Foundation (I.E.) and the NARSAD Young Investigator Award (S.F).

## Author contribution

I.E., R.S.D., J.H.K., and S.J.F. conceived and planned the experiments. S.E.H. analyzed the PET data with input from R.E.C., S.J.F., and M.N. S.E.H. and D.S. analyzed the MRI data. R.H.P. provided statistical expertise. G.S. assisted with recruitment. N.D. oversaw recruitment and scanning. M.T.D. assisted with screening. D.M. and G.A.A. provided medical expertise. N.N. was responsible for radiochemistry. S.E.H. wrote the manuscript in consultation with I.E. All authors helped shape the research, analysis, and manuscript.
