## [Peer Review File · Nature Communications]

Reviewers' comments:

Reviewer #1 (Remarks to the Author):

This is a very interesting study using an indirect measure of synaptic density in patients with MDD and PTSD. The translational relevance of the study is high and the experiment was carried out by an established group of investigators. The investigators take an interesting transdiagnostic approach, examining the SV2A marker in MDD, PTSD and comorbid, as well as HC. The sample sizes are small with "the clinical group" comprised of 11 individuals with MDD, 5 with PTSD and 10 with comorbid MDD and PTSD, which is a limitation.

The main finding is that SV2A density was negatively correlated with severity of depressive symptoms clinical subjects in dlPFC ($r=-0.633$, $p=0.001$), ACC ($r=-0.634$, $p=0.001$) and hippocampus ($r=-0.487$, $p=0.012$). As noted below, it was not immediately clear to me these findings are the results of a prior hypothesis testing or if the study is meant to be exploratory/hypothesis generating. This is an important point in the context of rigor. Either approach is reasonable, but it is incumbent upon the authors to present the analyses in light of one of the two approaches, I believe. The supportive analyses related to group effects of fMRI are interesting and appropriately presented as exploratory. Overall this is a very interesting paper with some potential limitation related to sample size and lack of clarity about pre-specified features of the statistical plan.

I have a few specific comments below.

1. Intro – maybe a minor quibble, but the paper starts with, "the human brain displays remarkable neuroplasticity in response to stress...Persisting high levels of stress or repeated exposure to extreme stress results in loss of synapses in circuits underlying affective and cognitive processes¹." But the data the authors are referring to is in rodent models, and I do not know if we know what happens re synaptic plasticity in humans under stress.

2. Intro, last paragraph, no specific hypotheses are stated. Authors refer to an exploratory approach, which is OK but should be consistent throughout. For example in the Results there is finding presented as primary of correlations between SV2A and a few select brain regions (where they defined a prior prior to data collection?) Then there is what is presented in the results as 'exploratory' but it is unclear if the authors are using these terms in the technical/statistical sense or just general language to indicate what they feel is most important. Since these terms do have technical meanings in this context, I suggest using the terms in that way.

3. Unclear if the p values presented in results section (e.g., for correlations between SVA and severity in 3 different regions, Fig 1) are corrected for multiple comparisons.

4. Methods – There is no explicit statement of hypothesis testing, sample size determination and the like. It is not explicitly stated if the hypotheses and the analytic plan for this experiment as presented was formulated before the data was collected. This is important in evaluating the results and in particular the risk of type II error. It is also important for general transparency. It would be helpful if the accompanying research protocol were available.

Reviewer #2 (Remarks to the Author):

This is an impressive study by a world-class imaging group. This is very novel work but there are concerns relevant to data interpretation, although it is important to note that the authors did well to address some key concerns through Supplemental data and the Discussion section. Example concerns: [1] need for further consideration of why it may be and/or why it might not be

appropriate to combine MDD, PTSD and MDD+PTSD in this first study; [2] limited samples sizes that include the number of "pure" PTSD (n=5), sizes of the LS and HS groups, and only a subset of HC with fMRI; [3] significantly different unlabeled mass doses administered across groups; [4] no correction for non-displaceable distribution volume of radiotracer; and [5] the use of "[C-11]UCB-J SV2A VT" and "synaptic density" as equivalent terms (e.g., axis labels). In reference 53 (Fig 1D), it appears that the correlation between SV2A VT and SV2A OD is $r^2=0.31$. How might this translate to the present work - is this lower correlation a concern?

Despite the authors' efforts, some of the work seems a bit exploratory (as noted by the authors) and larger sample sizes and further investigation are likely needed to establish the major findings. A few additional comments/questions remain. Did the authors observe any input function differences across the groups, e.g., did in vivo radiotracer metabolism differ across groups? Could gray matter atrophy impact data interpretation (beyond PVC for the PET and in terms of the MRI ICD metric)? Were regional gray matter volumes compared across groups or considered in the statistical analysis? The authors should describe UCB-J SV2A VT and VT/fp test-retest variability. Which outcome is considered more accurate or reliable, VT or VT/fp? Has outcome reliability been assessed yet in the relevant patient groups?

Overall, it is very exciting novel research.

Methods

- VT (line 423) also includes blood volume?
- Is the PVC method related to other methods more commonly applied in the past (e.g. GTM)? Please offer more clarification.
- fMRI: how many functional volumes were acquired? Were the PET data acquisition parameters listed (e.g dynamic acquisition)? Please provide further evidence that the parametric data were well fit.
- how does 5 mm fMRI resolution relate to PET resolution?

Results

- Scatter graphs would be more informative, if different symbols or colors indicated HC and/or patient sub-group membership
- It would be more informative to show scatter graphs rather than bar graphs when sample sizes are small
- Please list sample sizes in figure legends, when possible
- Why not show all results on the same y-axis scale? Please include "SV2A" on y-axis scale, although it really is "[C-11]UCB-J VT"
- Fig, 2b: Image planes appear to capture different anatomy for the 3 subjects (not spatially normalized?)
- Fig, 5: Graphs E and F are less convincing in terms of a linear relationship

Discussion

- line 226, ". . . exploratory analysis of additional brain regions points to a global effect." Were these data shown?
- The discussion is long and sometimes speculative (e.g., the paragraph on sleep, line 229) - consider shortening and further focusing the Discussion
- It is not always clear how the individuals in the current study (in a major depressive episode at screening) compare to subjects studied in some of the literature findings cited
- Line 305, the word "Second" may be missing here
- Line 322 The third aspect of the limitation discussion could be more clearly presented/described

Supplemental Data

- Table S3: Thalamus values appear to be missing

Reviewer #3 (Remarks to the Author):

The authors investigated SV2A with the radioligand [(11)C]UCB-J in healthy controls and an unmedicated clinical sample comprising subjects with either MDD, PTSD, or with comorbid MDD and PTSD. Both groups were matched for smoking and the ratio of male to female participants. In addition to PET imaging, functional brain connectivity was measured in the clinical sample and a fraction of the healthy controls with fMRI. Participants received a clinical assessment and computerized cognitive tasks.

The full clinical sample and the controls did not differ in SV2A VT. In the clinical sample increasing depressive symptom load corresponded to decreasing SV2A VT. When split into subjects with high and low depressive symptom load based on their HAMD score the high load group displayed lower SV2A VT than controls in the DLPFC, ACC, and hippocampus.

The full clinical sample displayed reduced functional connectivity of the DLPFC. Lower DLPFC SV2A VT corresponded to a higher DLPFC-PCC connectivity. Higher DLPFC-PCC connectivity corresponded to higher depressive symptom load and lower performance in a working memory task.

This carefully conducted investigation uses a novel radiotracer and presents the first in vivo evidence for altered SV2A VT in MDD. It bears a high potential impact for the field and would be of significant interest to the readership of this journal. Consider the points below to improve readability and address some important points in discussing the exciting findings of this study.

Major points

Stratification for symptom severity resulted in an uneven distribution of diagnoses across groups, with no subjects with PTSD without comorbid MDD in the high depressive symptom load group. This could be emphasized more clearly in the text.

The findings on the relation between depressive symptoms and SV2A VT are impressive. An additional aspect of the results would be that no relation was found between SV2A VT and symptoms of PTSD, or anxiety symptoms in general. Consistently, the very informative explorative analysis involving SV2A VT and the factorial structure of HAMD, revealed that decreased SV2A VT corresponds to higher scores on the depression, insomnia, and somatic, but not on the anxiety factor of HAMD. Consider expanding on this in the discussion.

Minor points

Information on the sensitivity and specificity of [(11)C]UCB-J, such as on page 13, lines 326-328, is central for understanding and appreciating the results. Therefore, consider moving it from the discussion to the introduction and extending it.

In the introduction the authors state that "binding to SV2A provides an estimate of the density of nerve terminals" (p. 4, line 94). However, in the results and discussion they use the terms "SV2A density" and "SV2A VT" synonymously, whereas in the figures they only use "SV2A VT" / "VT". Consider using "SV2A VT" throughout, or explicitly stating that alterations in SV2A VT indicate impaired SV2A density.

The discussion is interesting and informative but rather long and not very focussed. Consider shortening the text on page 10 (inflammation, HPA axis, mTORC1) to topics directly related to the specific methods and results of the present investigation.

The authors report a comparison of SV2A VT ACC-HIP correlation between subjects with high and low depressive symptom load on page 7, lines 160-165 (figure 4), and discuss it on page 9, lines 217-228. This analysis is refreshing, informative, and deserves more highlighting. Analyses on the cross-regional signal pattern in PET studies are highly underrepresented in spite of their potential to open new perspectives – they deserve more attention and appreciation.

Reviewer comments, author responses and manuscript changes (NCOMMS-18-32857-T)

We are grateful to each reviewer for the attentive reading of our work and their constructive comments. The comments are encouraging and all reviewers agree that the presented research is exciting and novel. Below is a point by point response to the reviewers' comments. We have carried out additional data analyses and updated the manuscript according to the reviewers' comments, and believe this makes for an improved manuscript.

Reviewer #1

This is a very interesting study using an indirect measure of synaptic density in patients with MDD and PTSD. The translational relevance of the study is high and the experiment was carried out by an established group of investigators. The investigators take an interesting transdiagnostic approach, examining the SV2A marker in MDD, PTSD and comorbid, as well as HC. The sample sizes are small with "the clinical group" comprised of 11 individuals with MDD, 5 with PTSD and 10 with comorbid MDD and PTSD, which is a limitation.

The main finding is that SV2A density was negatively correlated with severity of depressive symptoms clinical subjects in dlPFC ($r=-0.633$, $p=0.001$), ACC ($r=-0.634$, $p=0.001$) and hippocampus ($r=-0.487$, $p=0.012$). As noted below, it was not immediately clear to me these findings are the results of a prior hypothesis testing or if the study is meant to be exploratory/hypothesis generating. This is an important point in the context of rigor. Either approach is reasonable, but it is incumbent upon the authors to present the analyses in light of one of the two approaches, I believe. The supportive analyses related to group effects of fMRI are interesting and appropriately presented as exploratory. Overall this is a very interesting paper with some potential limitation related to sample size and lack of clarity about pre-specified features of the statistical plan.

I have a few specific comments below.

Comment 1: Intro – maybe a minor quibble, but the paper starts with, "the human brain displays remarkable neuroplasticity in response to stress...Persisting high levels of stress or repeated exposure to extreme stress results in loss of synapses in circuits underlying affective and cognitive processes¹." But the data the authors are referring to is in rodent models, and I do not know if we know what happens re synaptic plasticity in humans under stress.

Response: The reference cited (Duman et al. 2016¹) refers to both clinical and preclinical studies. It is well established that the human brain demonstrates neuroplastic changes in response to stress. For example, imaging studies show stress-induced functional changes in stress paradigms, and both functional and structural changes have been consistently shown in chronic stress disorders such as MDD and PTSD. We acknowledge that our statement that persisting levels of stress results in loss of synapses is, whilst directly supported by preclinical work, only indirectly supported by human studies (e.g. post-mortem studies in MDD). We have therefore changed the second sentence to "persisting high levels of stress are **thought to** result in loss of synapses in circuits...".

Comment 2: Intro, last paragraph, no specific hypotheses are stated. Authors refer to an exploratory approach, which is OK but should be consistent throughout. For example in the Results there is finding presented as primary of correlations between SV2A and a few select brain regions (where they defined a prior to data collection?) Then there is what is presented in the results as 'exploratory' but it is unclear if the authors are using these terms in the technical/statistical sense or just general language to indicate what they feel is most important. Since these terms do have technical meanings in this context, I suggest using the terms in that way.

Response: We have restructured the introduction to clarify that investigating the relationship between depression severity and synaptic density in a transdiagnostic sample was the primary focus of the study. We have also added our primary hypothesis to the end of the introduction:

“Our primary hypothesis was that synaptic density, as measured by [¹¹C]UCB-J PET, would be negatively associated with severity of depressive symptoms transdiagnostically”.

The regions were selected a priori based on existing research – the introduction outlines this research and justifies our a priori choice of regions. We also state this in the last paragraph of the introduction.

We have now clarified the primary and secondary analyses (see the data analysis section of the methods – pg. 19). Primary analyses included the transdiagnostic association between depression severity and synaptic density; and the group differences in synaptic density. Secondary analyses included the MRI results and the associations between synaptic density, cognitive function and the symptom clusters. Findings from the primary analyses were corrected for multiple comparisons. The secondary analyses relating to cognitive function and symptom clusters were not corrected for multiple comparisons, which is stated in the methods. It is important to note that although exploratory, the fMRI data were processed according to current best practices for multiple comparisons correction due to the high risk of type 1 error in fMRI research².

Comment 3: Unclear if the p values presented in results section (e.g., for correlations between SVA and severity in 3 different regions, Fig 1) are corrected for multiple comparisons.

Response: These correlations survived correction for multiple comparisons, which is stated in the results (pg. 5).

Comment 4: There is no explicit statement of hypothesis testing, sample size determination and the like. It is not explicitly stated if the hypotheses and the analytic plan for this experiment as presented was formulated before the data was collected. This is important in evaluating the results and in particular the risk of type II error. It is also important for general transparency. It would be helpful if the accompanying research protocol were available.

Response: We have added our primary hypothesis to the introduction and have outlined the primary and secondary analyses in the methods. The plan to adopt a transdiagnostic approach was formulated prior to data collection, and investigating the transdiagnostic association between synaptic density and depression severity was the primary aim of the study. Given that this is the first study of its kind there was no existing data on synaptic density and depression severity, and a precise power calculation was not possible. However, with anticipated large effect sizes based on pilot data and preclinical data on synaptic loss in models of depression, we determined that a clinical sample size of 24 would be sufficiently powered to detect a statistically significant (i.e., $p < 0.05$) association between synaptic density and depressive symptoms. The protocol is not available for distribution as it incorporates several other studies and other confidential information.

Reviewer #2

This is an impressive study by a world-class imaging group. This is very novel work but there are concerns relevant to data interpretation, although it is important to note that the authors did well to address some key concerns through Supplemental data and the Discussion section. Example concerns:

Comment 1: Need for further consideration of why it may be and/or why it might not be appropriate to combine MDD, PTSD and MDD+PTSD in this first study;

Response: Consistent with emerging theoretical frameworks in psychiatry (e.g., NIMH Research Domain Criteria Project) we intentionally employed a transdiagnostic, dimensional approach instead of a categorical approach. Diagnostic categories are inherently heterogeneous; a dimensional approach aims to understand the neurobiology underlying symptoms, which may lead to the discovery of targets and novel treatments that are effective for specific symptoms, in this case, depression. Indeed, depressive symptoms are common in PTSD, which is often comorbid with MDD, yet current antidepressants are often ineffective. We believe our transdiagnostic approach is a strength of this study, and that demonstrating an association between SV2A V_T , which is indicative of synaptic density, and depressive symptoms in a transdiagnostic sample of individuals with MDD and/or PTSD in a first study highlights the potential importance of synaptic plasticity in relation to depressive symptoms specifically. However, we acknowledge that larger PET studies investigating synaptic density in distinct categorical disorders are vital in further unravelling the role of synaptic density in relation to psychiatric symptoms, and have specifically addressed this in the Discussion.

Comment 2: Limited samples sizes that include the number of “pure” PTSD (n=5), sizes of the LS and HS groups, and only a subset of HC with fMRI;

Response: We acknowledge that the number of individuals with ‘pure’ PTSD is relatively small. However, given the transdiagnostic approach, we were able to detect a significant association between synaptic density and symptoms of depression. We would also like to highlight that 26 individuals with psychiatric illness (and 21 healthy controls; 47 subjects in total) is relatively large for a PET study. Further, all 26 individuals were medication-free, which is rare to see in a PET study of such size. Given that the study is sufficiently powered to detect a significant association between synaptic density and depressive symptoms transdiagnostically and that this is the first study of its kind, we believe that despite the relatively small samples of each diagnostic category alone, these findings add important and novel insight into the neurobiological mechanisms underlying symptoms of depression – symptoms that span multiple psychiatric and neurological disorders that are often inadequately treated. Saying this, as stated previously, we do acknowledge that larger PET studies investigating synaptic density in discrete categorical disorders are necessary to determine whether there are distinct synaptic density patterns across disorders, and we have highlighted this in the Discussion.

In relation to not having fMRI data in the full HC sample, we have highlighted this in the Discussion. Furthermore, we took several steps to improve power. For example, a ROI-based approach based on the PET findings was used to examine global connectivity in the dlPFC - indeed this finding replicates previous studies showing abnormal PFC connectivity in MDD and PTSD. Further, we combined PET and fMRI modalities to explore a relationship between lower synaptic density and a ‘blurring’ of 2 typically opposed networks in the clinical sample, which provides novel insight into the possible molecular underpinnings of network dysfunction.

Comment 3: Significantly different unlabelled mass doses administered across groups;

Response: Based on our previously reported validation study³ (<https://journals.sagepub.com/doi/abs/10.1177/0271678X17724947>), the mass doses in this study were determined to occupy <1% of receptors. Further, there was no observed correlation between injected mass and V_T . We are therefore confident that the different mass doses have not substantially impacted our results. It must also be noted that any effect of a lower mass dose in high

severity clinical group would have resulted in an *underestimation* of our observed group difference. This is because lower mass is associated with higher specific activity. Therefore, any effect of lower mass would have resulted in higher V_T and thus would have minimized the difference between high severity patients and controls. We have addressed this issue in the Discussion (pg.12).

Comment 4: No correction for non-displaceable distribution volume of radiotracer

Response: Given the observed difference in centrum semiovale V_T , and that MDD and PTSD have been associated with white matter pathology, we determined that V_T was the more appropriate outcome measure over BP_{ND} . V_T has been previously validated as a reliable and accurate outcome measure for assessing synaptic density using [^{11}C]UCB-J^{3,4}. Further, the non-displaceable uptake is very small compared to the grey matter uptake, especially after PVC and so nearly all of the grey matter V_T value represents specific binding. We address this in the Discussion (pg. 11).

Comment 5: The use of “[C-11]UCB-J SV2A VT” and “synaptic density” as equivalent terms (e.g., axis labels).

Response: Thank you for pointing this out - we have changed all axis labels to [^{11}C]UCB-J V_T and use [^{11}C]UCB-J V_T consistently throughout the results. We explicitly state that [^{11}C]UCB-J V_T is indicative of SV2A (synaptic) density, as validated previously^{3,4}.

Comment 6: In reference 53 (Fig 1D), it appears that the correlation between SV2A VT and SV2A OD is $r^2=0.31$. How might this translate to the present work - is this lower correlation a concern?

Response: The relatively low correlation between SV2A OD and SV2A V_T for the GM was due to the relative low difference in SV2A signal across the GM regions. Further, we’d like to highlight that Western blot quantification is semi-quantitative and the relatively small sample size for V_T (9 GM regions in one baboon). Considering the low sample size, it is important to also consider the high correlation between SV2A OD and SYN OD (direct comparison with the same method) and the good correlation between V_T and the SV2A density (B_{max}) from the homogenate binding assay (a more accurate assay for SV2A). Further, blockade of [^{11}C]UCB-J binding by levetiracetam confirms SV2A density *in vivo*. We are thus comfortable that V_T relates to SV2A density and that SV2A and SYN expression was highly correlated across the brain regions.

Comment 7: Did the authors observe any input function differences across the groups, e.g., did *in vivo* radiotracer metabolism differ across groups?

Response: We observed no between-group differences in plasma parent fraction at either 30 or 60 minutes post-injection. Further, there was no difference in the metabolite-corrected input function (40-60 minutes post-injection) across groups. We have added this to the manuscript and now report these data in SI (Table S4; shown below). Note that if there was a group difference in the tracer availability, the use of kinetic modelling would have provided a correction for this effect.

Table S4. Parent fraction and metabolite-corrected input function values across groups

	HC group (n=21)	Low severity group (n=14)	High severity group (n=12)	HC vs. low severity p -value	HC vs. high severity p -value
Parent fraction 30 min (%)	28 (7)	23 (7)	27 (8)	0.08	0.81
Parent fraction 60 min (%)	25 (5)	22 (6)	23 (5)	0.14	0.39
Metabolite-corrected input function 40-60 min (SUV)	0.25 (0.05)	0.22 (0.05)	0.24 (0.06)	0.12	0.27

Values are presented as mean (SD)

Comment 8: Could gray matter atrophy impact data interpretation (beyond PVC for the PET and in terms of the MRI ICD metric)?

Response: It is unlikely that gray matter atrophy could account for the ICD findings. As described in the response below, we did not detect any group differences in gray matter volume for the a priori ROIs. In our two previous papers using ICD in MDD and PTSD, we showed that regions of group differences in ICD were not the same regions as group differences in gray matter volume^{5,6}. Further, we have shown that the correlation between gray matter volume and ICD is not significantly correlated across the gray matter in MDD⁵. Finally, ICD involves several steps to account for potential confounds related to brain size, include high resolution non-linear registrations, and within subject normalization of connectivity values.

Comment 9: Were regional gray matter volumes compared across groups or considered in the statistical analysis?

Response: Regional gray matter volumes were calculated using tensor-based morphometry (TBM) as in our previous studies^{5,6}. The determinant of the Jacobian of the deformation field generated from a high resolution non-linear transformation into MNI common space was used to quantify local volume differences between the registered images and the template⁷. This metric provided an estimate of voxel-wise volume changes for all transformed images with respect to the group averaged template and was used for further analysis. As shown in Response Table 1, below, there were no differences in gray matter volume for these regions, suggesting that there were no differences in gray matter volume for the three regions of interest. As such, we did not consider gray matter volume in any further statistical analysis.

Table 1. TBM results

	HC group	Low severity group	High severity group	p -value (low severity vs HC)	P -value (high severity vs. HC)
ACC	32.41±23.52	-38.85±35.02	-29.91±19.57	0.16	0.37
Hippocampus	-0.07±9.24	0.64±15.51	-2.71±30.68	0.88	0.44
Amygdala	8.59±22.61	2.77±51.53	5.14±15.00	0.70	0.87

Values are presented as mean±SD. *P*-values are obtained from independent-samples t-tests.

Comment 10: The authors should describe UCB-J SV2A VT and VT/*f_p* test-retest variability. Which outcome is considered more accurate or reliable, VT or VT/*f_p*?

Response: Our previous paper³ describes test-retest variability of [¹¹C]UCB-J *V_T* and *V_T/f_p*. The test-retest reproducibility of [¹¹C]UCB-J *V_T* was exceptionally good, TRV characteristics were virtually identical for ROI and voxel-based analyses. For voxel-based analysis, the regional mean TRV and aTRV for *V_T* were -0.7±5.2% and 4.2±2.6. Correction for *f_p* worsened the TRV and ICC. Indeed the ICC was above 0.6 (the commonly used as a threshold) for *V_T/f_p*, but <0.6 for *V_T*. Therefore, we chose *V_T* as our primary outcome measure, but report *V_T/f_p* in SI for completeness. We have added a statement on test-retest variability of these outcomes to the methods section (under 'PET image analysis').

Comment 11: Has outcome reliability been assessed yet in the relevant patient groups?

Response: Although test-retest studies are not typically performed in psychiatric populations, we conducted repeat scans in two adults with MDD and observed test-retest reliability similar to

controls [$r=1.00$ and no significant change in outcome measure in HIP ($t=0.23$, $p=0.86$) or dlPFC ($t=1.16$, $p=0.45$)].

Comment 12: Overall, it is very exciting novel research.

Response: Thank you, we agree!

Comment 13: VT (line 423) also includes blood volume?

Response: Yes, V_T does include blood volume. However, as we showed previously³, inclusion of the blood volume fraction into the 1T model fit did not affect regional K_1 or V_T values (supplementary figures 1 and 2). This is because blood volume in the brain is around 5% and the corresponding radioactivity low compared to brain tissue radioactivity, due to the very high binding of [¹¹C]UCB-J in brain tissue. We have added this to the description of V_T in the methods.

Comment 14: Is the PVC method related to other methods more commonly applied in the past (e.g. GTM)? Please offer more clarification.

Response: We applied the Muller-Gartner PVC method to our data, as this allows for voxel-wise PVC in line with our image analysis methods (our primary outcome measure was V_T computed parametrically). In contrast, the geometric transfer matrix (GTM) method is applied in studies using ROI analysis. The Muller-Gartner method is an extension of the voxel-based Videen method, where the MRI is segmented into GM, WM and CSF in order to obtain more accurate GM values⁸. We have now specified that we used the Muller-Gartner method in the manuscript.

Comment 15: fMRI: how many functional volumes were acquired?

Response: There were 340 volumes acquired per run, and each subject had 2 runs. This is detailed in the methods.

Comment 16: Were the PET data acquisition parameters listed (e.g dynamic acquisition)?

Response: Thank you for pointing this out - these have now been added to SI.

Comment 17: Further evidence that the parametric data were well fit.

Response: Our previous test-retest paper³ provides a comprehensive assessment of model fitting for [¹¹C]UCB-J data and indicates that the 1TCM is optimal, providing high precision and ease of production of parametric images. Further, V_T values obtained from the 1TCM from regional TAC based values agreed extremely well with values obtained using parametric maps ($R^2=0.99$). We have stated in the methods (pg. 16) that calculation of V_T parametrically using the 1TCM has been validated previously.

Comment 18: How does 5 mm fMRI resolution relate to PET resolution?

Response: We would like to clarify that the resolution of fMRI is 2.5mm isotropic. The PET data were obtained on the HRRT (the highest resolution human PET system available), which has a similar resolution to fMRI (2-3mm). Additionally, the fMRI and PET data were averaged across nodes of the same AAL atlas, minimizing any effects of resolution differences as both sets of data were reduced to the same dimensions.

Comment 19: Scatter graphs would be more informative, if different symbols or colors indicated HC and/or patient sub-group membership-

Response: Thank you for this suggestion – we agree this is more informative and have changed the bar graph in figure 2 (shown below) to a scatter graph with different symbols representing the clinical subgroups; which is detailed in the figure legend.

Figure 2. Lower [¹¹C]UCB-J V_T (SV2A density) in individuals with high severity depressive symptoms vs HC subjects

a. [¹¹C]UCB-J V_T in the dIPFC, ACC and hippocampus across groups. The low severity group (in green; n=14) consisted of individuals with HAMD-17 scores <14 and the high severity group (in blue; n=12) consisted of individuals with HAMD-17 scores ≥14. clinical subgroups are represented by different symbols; the low severity group consisted of MDD (n=5), PTSD (n=5) and MDD/PTSD (n=4) subjects; the high severity group consisted of MDD (n=6) and MDD/PTSD (n=6) subjects **b.** Representative axial, coronal and sagittal parametric images of [¹¹C]UCB-J PET (V_T) scans registered to MR images in MNI space from a HC subject (top), low severity subject (middle) and high severity subject (bottom).

Comment 20: Please list sample sizes in figure legends, when possible

Response: We have now added sample sizes to all relevant figures.

Comment 21: Why not show all results on the same y-axis scale? Please include “SV2A” on y-axis scale, although it really is “[C-11]UCB-J VT”

Response: We have changed the dIPFC and ACC graphs from figure 1 to the same axis but not for the hippocampus due to lower values. We have changed all graph axis titles to [¹¹C]UCB-J V_T.

Comment 22: Fig, 2b: Image planes appear to capture different anatomy for the 3 subjects (not spatially normalized?)

Response: We have normalised the images to MNI space and the same planes are now shown for all 3 subjects (see updated figure 2, shown above)

Comment 23: Fig, 5: Graphs E and F are less convincing in terms of a linear relationship

Response: As stated, the cognitive results are exploratory and we acknowledge that replication is necessary to establish a linear relationship.

Comment 24: Line 226, “. . . exploratory analysis of additional brain regions points to a global effect.” Were these data shown?

Response: Yes, these data are shown in SI (Tables S1, S2 and S3) and are stated in the results.

Comment 25: The discussion is long and sometimes speculative (e.g., the paragraph on sleep, line 229) - consider shortening and further focusing the Discussion.

Response: Thank you for bringing this to our attention. The Discussion has now been shortened (for example, the paragraph on sleep and somatic symptoms has been shortened; and the paragraph on mechanisms leading to synaptic density changes has been removed). We believe the Discussion is now more concise and focussed.

Comment 26: It is not always clear how the individuals in the current study (in a major depressive episode at screening) compare to subjects studied in some of the literature findings cited

Response: In the cited studies, MDD and PTSD patients were actively symptomatic at the time of study. However, due to the heterogeneity in psychiatric disorders, severity, duration of illness and age of onset were varied across studies.

Comment 27: Line 305, the word “Second” may be missing here

Response: Thank you, this has been added.

Comment 28: Line 322 The third aspect of the limitation discussion could be more clearly presented/described

Response: This has been amended and is now described more clearly.

Comment 29: Table S3: Thalamus values appear to be missing

Response: This has now been added.

Reviewer #3

The authors investigated SV2A with the radioligand [(11)C]UCB-J in healthy controls and an unmedicated clinical sample comprising subjects with either MDD, PTSD, or with comorbid MDD and PTSD. Both groups were matched for smoking and the ratio of male to female participants. In addition to PET imaging, functional brain connectivity was measured in the clinical sample and a fraction of the healthy controls with fMRI. Participants received a clinical assessment and computerized cognitive tasks.

The full clinical sample and the controls did not differ in SV2A VT. In the clinical sample increasing depressive symptom load corresponded to decreasing SV2A VT. When split into subjects with high and low depressive symptom load based on their HAMD score the high load group displayed lower SV2A VT than controls in the DLPFC, ACC, and hippocampus.

The full clinical sample displayed reduced functional connectivity of the DLPFC. Lower DLPFC SV2A

VT corresponded to a higher DLPFC-PCC connectivity. Higher DLPFC-PCC connectivity corresponded to higher depressive symptom load and lower performance in a working memory task. This carefully conducted investigation uses a novel radiotracer and presents the first in vivo evidence for altered SV2A VT in MDD. It bears a high potential impact for the field and would be of significant interest to the readership of this journal. Consider the points below to improve readability and address some important points in discussing the exciting findings of this study.

Comment 1: Stratification for symptom severity resulted in an uneven distribution of diagnoses across groups, with no subjects with PTSD without comorbid MDD in the high depressive symptom load group. This could be emphasized more clearly in the text.

Response: This is an important point – we have now amended figure 2 to indicate the breakdown of patient subgroups in each severity group – and have specified the numbers of each group in the text of the figure legend. Further, we have listed this as a specific limitation in the Discussion, and state that larger studies investigating synaptic density within each diagnostic subgroup are necessary going forward. It is also important to point out that MDD comorbidity in a PTSD diagnosis is a sign of more severe PTSD⁹. Thus, it is expected that PTSD group without MDD would not be in the high severity sample.

Comment 2: The findings on the relation between depressive symptoms and SV2A VT are impressive. An additional aspect of the results would be that no relation was found between SV2A VT and symptoms of PTSD, or anxiety symptoms in general. Consistently, the very informative explorative analysis involving SV2A VT and the factorial structure of HAMD, revealed that decreased SV2A VT corresponds to higher scores on the depression, insomnia, and somatic, but not on the anxiety factor of HAMD. Consider expanding on this in the discussion.

Response: Thank you for this suggestion – we agree this needed highlighting further and have done so on page 11 of the Discussion.

Comment 3: Information on the sensitivity and specificity of [(11)C]UCB-J, such as on page 13, lines 326-328, is central for understanding and appreciating the results. Therefore, consider moving it from the discussion to the introduction and extending it.

Response: This makes sense thank you - we have highlighted some of the excellent imaging properties of [¹¹C]UCB-J in the introduction (pg.4).

Comment 4: In the introduction the authors state that “binding to SV2A provides an estimate of the density of nerve terminals” (p. 4, line 94). However, in the results and discussion they use the terms “SV2A density” and “SV2A VT” synonymously, whereas in the figures they only use “SV2A VT” / “VT”. Consider using “SV2A VT” throughout, or explicitly stating that alterations in SV2A VT indicate impaired SV2A density.

Response: We have changed the axis labels to [¹¹C]UCB-J V_T as suggested by reviewer 2, and have made it clear that [¹¹C]UCB-J V_T is indicative of synaptic density.

Comment 5: The discussion is interesting and informative but rather long and not very focussed. Consider shortening the text on page 10 (inflammation, HPA axis, mTORC1) to topics directly related to the specific methods and results of the present investigation.

Response: Whilst these topics are relevant and interesting to discuss, we agree the Discussion could be more focussed and have now made the Discussion more concise (the length has been cut by 15%).

Comment 6: The authors report a comparison of SV2A VT ACC-HIP correlation between subjects with high and low depressive symptom load on page 7, lines 160-165 (figure 4), and discuss it on page 9, lines 217-228. This analysis is refreshing, informative, and deserves more highlighting. Analyses on the cross-regional signal pattern in PET studies are highly underrepresented in spite of their potential to open new perspectives – they deserve more attention and appreciation.

Response: Thank you for this positive feedback. We agree these types of analyses are important and have highlighted this finding in the first paragraph of the Discussion.

References

- 1 Duman, R. S., Aghajanian, G. K., Sanacora, G. & Krystal, J. H. Synaptic plasticity and depression: new insights from stress and rapid-acting antidepressants. *Nature medicine* **22**, 238 (2016).
- 2 Eklund, A., Nichols, T. E. & Knutsson, H. Cluster failure: Why fMRI inferences for spatial extent have inflated false-positive rates. *Proceedings of the National Academy of Sciences* **113**, 7900-7905, doi:10.1073/pnas.1602413113 (2016).
- 3 Finnema, S. J. *et al.* Kinetic evaluation and test–retest reproducibility of [11C] UCB-J, a novel radioligand for positron emission tomography imaging of synaptic vesicle glycoprotein 2A in humans. *Journal of Cerebral Blood Flow & Metabolism*, 0271678X17724947 (2017).
- 4 Finnema, S. J. *et al.* Imaging synaptic density in the living human brain. *Science translational medicine* **8**, 348ra396-348ra396 (2016).
- 5 Scheinost, D. *et al.* Multimodal Investigation of Network Level Effects Using Intrinsic Functional Connectivity, Anatomical Covariance, and Structure-to-Function Correlations in Unmedicated Major Depressive Disorder. *Neuropsychopharmacology* **43**, 1119-1127, doi:10.1038/npp.2017.229 (2018).
- 6 Holmes, S. E. *et al.* Cerebellar and prefrontal cortical alterations in PTSD: structural and functional evidence. *Chronic stress (Thousand Oaks, Calif.)* **2**, doi:10.1177/2470547018786390 (2018).
- 7 Hua, X. *et al.* Tensor-based morphometry as a neuroimaging biomarker for Alzheimer's disease: an MRI study of 676 AD, MCI, and normal subjects. *Neuroimage* **43**, 458-469, doi:10.1016/j.neuroimage.2008.07.013 (2008).
- 8 Erlandsson, K., Buvat, I., Pretorius, P. H., Thomas, B. A. & Hutton, B. F. A review of partial volume correction techniques for emission tomography and their applications in neurology, cardiology and oncology. *Physics in Medicine & Biology* **57**, R119 (2012).
- 9 Post, L. M., Zoellner, L. A., Youngstrom, E. & Feeny, N. C. Understanding the relationship between co-occurring PTSD and MDD: Symptom severity and affect. *Journal of Anxiety Disorders* **25**, 1123-1130, doi:<https://doi.org/10.1016/j.janxdis.2011.08.003> (2011).

REVIEWERS' COMMENTS:

Reviewer #1 (Remarks to the Author):

All of my comments have been addressed.

Thank you

Reviewer #3 (Remarks to the Author):

My concerns have been sufficiently addressed by the authors in the revised version of the manuscript.

Gregor Hasler